# COVID-19 related posttraumatic stress disorder in children and adolescents in Saudi Arabia

**Mohamed H. Sayed**[1,2☺], **Moustafa A. Hegazi** [1,3☺] *****, **Mohamed S. El-Baz**[1,2‡], **Turki S. Alahmadi**[1,4‡], **Nadeem A. Zubairi**[1‡], **Mohammad A. Altuwiriqi**[1‡], **Fajr A. Saeedi**[1‡], **Ali F. Atwah**[1‡], **Nada M. Abdulhaq**[1‡], **Saleh H. Almurashi**[5‡]

**1** Department of Pediatrics, Faculty of Medicine in Rabigh, King Abdulaziz University, Jeddah, Saudi Arabia,
**2** Department of Pediatrics, Faculty of Medicine, Cairo University, Cairo, Egypt, **3** Department of Pediatrics, Mansoura University Children's Hospital, Mansoura, Egypt, **4** Department of Pediatrics, King Abdulaziz University Hospital, Jeddah, Saudi Arabia, **5** Faculty of Medicine in Rabigh, King Abdulaziz University, Jeddah, Saudi Arabia

☺ These authors contributed equally to this work.
‡ These authors also contributed equally to this work.
* mhhegazi712003@yahoo.co.uk, mahhassan@kau.edu.sa

**Data Availability Statement:** All relevant data are within the manuscript and its Supporting Information files. Additionally, the raw data SPPS file was made available and deposited to figshare

## Abstract

### Introduction

The COVID-19 pandemic resulted in quarantine/lockdown measures in most countries. Quarantine may create intense psychological problems including post-traumatic stress disorder (PTSD) especially for the vulnerable critically developing children/adolescents. Few studies evaluated PTSD associated with infectious disasters but no Saudi study investigated PTSD associated with COVID-19 in children/adolescents. This study was undertaken to screen for PTSD in children/adolescent in Saudi Arabia to identify its prevalence/risk factors during COVID-19 pandemic and its quarantine.

### Methods

A cross-sectional survey was conducted after 2 months form start of quarantine for COVID-19 pandemic utilizing the original English version and an Arabic translated version for the University of California at Los Angeles Brief COVID-19 Screen for Child/Adolescent PTSD that can be parent-reported or self-completed by older children/adolescents. Participants (Saudi citizens/non-Saudi residents) were approached online via social media.

### Results

Five hundred and thirty seven participants were enrolled. The participants were 262 boys and 275 girls with a mean age of 12.25±3.77 years. Symptoms of no, minimal, mild and potential PTSD were identified in 15.5%, 44.1%, 27.4% and 13.0% of children/adolescents, respectively. The age, gender, school grade, and residence were not predictive of PTSD symptoms. Univariate analysis of risk factors for PTSD revealed that work of a close relative around people who might be infected was significantly different between groups of PTSD

public data repository (DOI: 10.6084/m9.figshare.14565531).

**Funding:** The author(s) received no specific funding for this work.

**Competing interests:** The authors have declared that no competing interests exist.

symptoms, but this difference disappeared during multivariate analysis. Children/adolescents of Saudi citizens had significantly lower median total PTSD score than children/adolescents of expatriate families (p = 0.002).

## Conclusion

PTSD associated with the COVID-19 and its resultant quarantine shouldn't be overlooked in different populations as it is expected in a considerable proportion of children/adolescents with variable prevalence, risk factors and severity. Parents/healthcare providers must be aware of PTSD associated with COVID-19 or similar disasters, so, they can provide children/adolescent with effective coping mechanisms.

## Introduction

The COVID-19 that appeared in China in December, 2019, was officially recognized by the World Health Organization as a pandemic on January 30, 2020. This virus has, since then, spread to all countries with variable burden, severity, psychological and socioeconomic impacts [1].

In response to SARS-CoV-2 spread, lockdown measures have been implemented in most of the affected countries to stop/slow down further transmission of the deadly virus [2, 3]. Such disease containment measures may suppress the outbreak, but they can also adversely affect family values, rituals and norms, which otherwise protect and regulate family functioning in times of disasters [4].

Quarantine is a preventive measure to safeguard public health and is characterized by isolation of persons who have been exposed to an infection, restriction of their movements separating them from others, for a specified time period [5]. Quarantine is beneficial for the general public to contain and limit the spread of an infectious agent, but it may create intense emotional, psychological and financial problems for some individuals [6, 7]. Global pandemics, in fact, are known to have intensively affected the mental well-being of individuals and masses [8].

Quarantine is often a troublesome upsetting experience with dramatic effects for those who suffer it. It is usually associated with multiple stressors such as its long duration, fright of infection, frustration, monotony, separation from loved ones, loss of freedom, financial loss, inadequate basic supplies (water, food, etc..), stigmatization and rejection by other people in the neighborhood, inadequate information with uncertainty over disease status and inadequate clear guidelines for necessary actions [6].

The critically developing children/adolescents are among the most vulnerable groups for community-based attenuation measures to fight COVID-19. This can disturb the usual lifestyle of children with closure of schools, parks, and playgrounds that may cause confusion and florid mental distress. Children/adolescents have to face these changes and may manifest signs of hostility, impatience, and intolerance. This may result in provoking the already over stressed parents to cause physical and mental violence to such children [9]. Thus, children needing quarantine due to confirmed or suspected COVID-19 infection might need extra efforts to pacify their fear, anxiety, and other psychological problems [10, 11].

Massive fear of COVID-19 has created plenty of psychiatric problems across several categories of the community [12]. Posttraumatic stress disorder (PTSD) has emerged as an important psychiatric problem. Its severity has been found to be directly proportionate to the duration of quarantine [7, 13].

PTSD is an anxiety disorder that can occur in persons who have been exposed to a traumatic, violent or catastrophic event such as earthquakes, hurricanes or pandemics like COVID-19, or who witnessed such experience happening to some close relatives [14]. PTSD is characterized by ignoring stimuli associated with a traumatic experience, restoring the trauma event, and hyperarousal, such as increased attentiveness that can extend for more than a month following exposure to trauma. This usually leads to significant interference with individual's normal working and social life, including school performance in case of children. Children can suffer dramatic changes in their cognition and mood, presenting with problems of behavior, attention and ability to concentrate in the school environment [14, 15].

Few studies [7, 16] have evaluated PTSD associated with infectious disasters but no study from Saudi Arabia has investigated the effects of COVID-19 and its resultant long duration of quarantine on children/adolescents. Therefore, this study was undertaken to screen for symptoms associated with PTSD in children/adolescents in Saudi Arabia to identify the prevalence/magnitude and risk factors of PTSD as a consequence of COVID-19 pandemic and its long period of quarantine.

## Materials and methods

### Study design and selection of participants

A cross-sectional survey was designed utilizing Brief COVID-19 Screen for Child/Adolescent (BCSCA) PTSD developed by University of California at Los Angeles (UCLA) [17]. This survey was conducted in the Kingdom of Saudi Arabia (KSA) from 12[th] of March to 12[th] of June 2020, two months after start of quarantine for COVID-19 pandemic. Participants were randomly selected and approached by an electronic online form of BCSCA. Parents and children/adolescents with age range of 6–18 years were invited to answer this questionnaire if they were Saudi citizens or residents and experienced the COVID-9 quarantine. Participants living outside KSA or having children outside the specified age range of 6–18 years as well as incomplete questionnaires were excluded from this study.

### Ethical considerations

Ethical approval was obtained from the biomedical ethics unit of Faculty of Medicine of King Abdulaziz University. A written informed consent was not obtained from parents or participants under 18 years old as filling the required online questionnaire by participants was considered their consent for participation in this survey. Participants' identity and confidentiality of their responses were protected.

### Measuring tool

The UCLA-BCSCA for PTSD questionnaire was used which is a newly developed tool to specifically help in screening for COVID-19-related PTSD in children/adolescents. The UCLA-BCSCA is derived from the validated UCLA PTSD reaction index for the Diagnostic and Statistical Manual of Mental Disorders, 5th edition (DSM-5) and it is one of the most commonly used instruments for the evaluation of traumatized children/adolescents. The UCLA-BCSCA for PTSD includes an initial set of questions (e.g., *Have you or someone close to you become very sick or been in the hospital because of this illness*? *Has anyone close to you died because of this illness*? *Does someone close to you work around people who might have this illness*?), to briefly review the traumatic event and set the stage for the subsequent related questions. Thus, it assists the child in recalling details of the traumatic event. These initial questions are followed by an 11-item set of validated questions about the frequency of PTSD symptoms

in the past month (rated from 0 = none of the time to 4 = most of the time), including 4 symptom categories (Category B: Intrusion symptoms; questions 4, 7, 10, Category C: Avoidance symptoms; questions 1, 6, Category D: Negative conditions/mood symptoms; questions 5,8,9 and Category E: Arousal/Reactivity symptoms; questions 2,3,11) [18, 19].

The UCLA BCSCA assessment tool includes reaction index total scale score based on (DSM-5) PTSD diagnostic screener, with screener rating from 1–10 denoting minimal PTSD symptoms, from 11–20 denoting mild PTSD symptoms whereas rating of 21 or higher denotes potential PTSD and warrants further evaluation by full PTSD-reaction index assessment and triage.

## Questionnaire implementation and distribution

The UCLA-BCSCA is originally available in English and it was translated into Arabic, checked by two bilingual experts and used in a pilot study for Arabic speaking participants to detect if any amendments are required.

Both English and Arabic questionnaires were converted to Google forms, so participants can select to fill the most convenient questionnaire for them. Then, the links of both questionnaires were sent via social media, including WhatsApp's, Facebook and Twitter, to participants. Participants were allowed to send the questionnaire's link to their relatives and friends as a mean to increase the sample size.

The parental-reported/completed questionnaire was directed to parents of children who can fill the whole questionnaire including response to questions directed for their children after taking opinions and answers first from their eldest child. Furthermore, the parent could also enter the questionnaire link again to fill further questionnaire(s) for other children between 6–18 years. The whole questionnaire was self-completed by children/adolescents who could understand the questions without the help of their parents. However, if children/adolescents ask for the help of their parents to explain any question, parents were present beside them as interviewers to explain any raised question or issue.

## Sample size and study power

The sample size was calculated on the basis of the published data on the prevalence of PTSD in 30% of isolated or quarantined children according to parental reports associated with pandemic disasters and short quarantine periods during pandemic influenza H1N1 and utilizing a closely similar measuring tool (UCLA-PTSD-Reaction Index-Parent Version) [16]. An estimated sample size of 233 from the population of children/adolescents (6–18 years) in KSA, achieved 90% power to detect a difference of 10% between the hypothesized proportion of 30% and the alternative hypothesis that the maximum expected proportion is 40% with longer quarantine period using a two-sided, binomial hypothesis test with a target significance level <0.05. However, 537 complete responses could be collected which exceeded the required estimated sample size and extremely increased the power of the study to more than 99%. The sample size and study power were calculated by PASS software (Pass output in S1 Fig).

## Statistical analysis

Data were analyzed by SPSS version 25.0 (IBM corporation, Armonk, NY) after checking for completeness and inconsistencies. Data were scrutinized and double-checked before and after entry into SPSS program. Frequencies and percentages were used to represent categorical variables, whereas continuous variables were presented as mean, standard deviation (SD) and range. Kolmogorov Smirnov (KS) test was used to test data normality. The associations between qualitative data were compared by chi-square test. Kruskal Wallis test was used to test

the differences between medians of the total UCLA-BCSCA PTSD score of the studied variables. Significance was considered at P value <0.05.

## Results

Five hundred and thirty-seven participants including 494 Saudi citizens and 43 non-Saudi residents from 5 main regions of KSA completely filled the questionnaire. The participants were 262 (48.8%) boys and 275 (51.2%) girls with a mean and SD of 12.25 ± 3.77 years and a range from 7–18 years. Three hundred and twenty-eight parents (61.1%) had 3 or more children. The sociodemographic data of participants are presented in Table 1.

In the initial set of questions to explore for traumatic experience with COVID-19 (Table 2), the most frequently reported traumatic effects/events of COVID-19 on Saudi children/adolescents and their families/close relatives were the fears from work of close relative around people who might have COVID-19, followed by upsetting issues that happened to children/adolescents or their families because of COVID-19 in 29.8% and 9.5% of children/adolescents respectively (Table 2).

The most frequent upsetting issues were being away from some family members (because of isolation or lockdown in other country), home stay/isolation from outside world for long period and work/isolation of mother/father in health care center reported by 23.5%, 21.6%, and 9.8% of participants respectively (S1 Table). The death of any close relative/friend because of COVID-19 was only recorded by 21 participants (3.9%) with death of 15 close relatives (one sister, 2 brothers, 3 grandmothers, 3 cousins, 2 father's aunts, 2 father's cousins, and 2 uncle's wives) and 6 friends/neighbors.

A median score of 1 was recorded for question number 1 '*I try to stay away from people, places or things that remind me about what happened or what is still happening*' and question number 4 '*When something reminds me of what happened or is still happening, I get very upset, afraid or sad*' as well as for question number 6 '*I try not to think about or have feelings about what happened or is still happening*' (Table 3).

**Table 1. Sociodemographic characteristics of participants (n = 537).**

| Character | | n | (%) |
|---|---|---|---|
| **Nationality** | Saudi citizen | 494 | (92.0) |
| | Non-Saudi resident | 43 | (8.0) |
| **Region** | Central | 48 | 8.9 |
| | Western | 433 | 80.6 |
| | Eastern | 19 | 3.5 |
| | North | 23 | 4.3 |
| | South | 14 | 2.6 |
| **Number of children in family** | 1 | 76 | 14.1 |
| | 2 | 133 | 24.8 |
| | 3 | 103 | 19.2 |
| | 4 | 103 | 19.2 |
| | 5 | 64 | 11.9 |
| | >5 | 58 | 10.8 |
| **Children/adolescent gender** | Male | 262 | 48.8 |
| | Female | 275 | 51.2 |
| **School grade of child/adolescent** | Primary (grades1-6) | 259 | 48.2 |
| | Intermediate (grades 7–9) | 120 | 22.3 |
| | Secondary (grades 10–12) | 158 | 29.4 |

**Table 2. Traumatic effects/events of COVID-19 in Saudi children/adolescents and their families/close relatives.**

| Effect/Event | | n | (%) |
|---|---|---|---|
| **Illness of child/close relative because of COVID-19** | No | 517 | 96.3 |
| | Yes | 20 | 3.7 |
| **Quarantine of child/close relative because of COVID-19** | No | 512 | 95.3 |
| | Yes | 25 | 4.7 |
| **Positive SARS-CoV-2 of child/close relative** | No | 501 | 93.3 |
| | Yes | 36 | 6.7 |
| **Work of close relative around people who might have COVID-19** | No | 377 | 70.2 |
| | Yes | 160 | 29.8 |
| **Moving of a family member away from home because of COVID-19** | No | 509 | 94.8 |
| | Yes | 28 | 5.2 |
| **Death of any close relative/friend because of COVID-19** | No | 516 | 96.1 |
| | Yes | 21 | 3.9 |
| **Anything else happened to you or your family because of SARS-CoV-2** | No | 486 | 90.5 |
| | Yes | 51 | 9.5 |

Regarding the rating of PTSD symptoms, no PTSD symptom, minimal PTSD symptom, mild PTSD symptoms and potential PTSD were identified in 83 (15.5%), 237 (44.1%), 147 (27.4%) and 70 (13.0%) of participating children/adolescents respectively (Table 4).

Participating children/adolescents declared that symptom number 4 '*When something reminds me of what happened or is still happening, I get very upset, afraid or sad*' is the most

**Table 3. Values of individual symptoms and category symptoms of UCLA-BCSCA for PTSD.**

| Individual/Category symptoms | Median | IQR (Q25-Q75) | Min | Max |
|---|---|---|---|---|
| **Q1:** I try to stay away from people, places or things that remind me about what happened or what is still happening | 1.0 | 0.0–3.0 | 0 | 4 |
| **Q2:** I get upset easily, or get into arguments, or physical fights | 0.0 | 0.0–1.0 | 0 | 4 |
| **Q3:** I have trouble concentrating or paying attention | 0.0 | 0.0–1.0 | 0 | 4 |
| **Q4:** When something reminds me of what happened or is still happening, I get very upset, afraid or sad | 1.0 | 0.0–2.0 | 0 | 4 |
| **Q5:** I have trouble feeling happiness or love | 0.0 | 0.0–1.0 | 0 | 4 |
| **Q6:** I try not to think about or have feelings about what happened or is still happening | 1.0 | 0.0–2.0 | 0 | 4 |
| **Q7:** When something reminds me of what happened, I have strong feelings in my body like heart rate beats fast, my head aches or my stomach aches | 0.0 | 0.0–1.0 | 0 | 4 |
| **Q8:** I have thoughts like "I will never be able to trust other people" | 0.0 | 0.0–1.0 | 0 | 4 |
| **Q9:** feel alone even when I am around other people | 0.0 | 0.0–1.0 | 0 | 4 |
| **Q10:** I have upsetting thoughts, pictures or sounds of what happened or is still happening come into my mind when I don't want them to | 0.0 | 0.0–1.0 | 0 | 4 |
| **Q11:** I have trouble going to sleep, wake up often, or have trouble getting back to sleep | 0.0 | 0.0–1.0 | 0 | 4 |
| **Category B** (Intrusion symptoms-Q4,7,10) | 2.0 | 0.0–4.0 | 0 | 12 |
| **Category C** (Avoidance symptoms-Q 1,6) | 2.0 | 0.0–4.0 | 0 | 8 |
| **Category D** (Negative conditions/mood symptoms-Q5,8,9) | 1.0 | 0.0–4.0 | 0 | 12 |
| **Category E** (Arousal/Reactivity symptoms-Q2,3,11) | 2.0 | 0.0–4.0 | 0 | 12 |
| **Total score (11Q)** | 8.0 | 3.0–15 | 0 | 41 |

**Table 4. Rating of posttraumatic stress disorder symptoms.**

| Rating | Frequency (n) | % |
|---|---|---|
| 0: no PTSD symptom | 83 | 15.5 |
| 1–10 minimal PTSD symptom | 237 | 44.1 |
| 11–20 mild PTSD symptoms | 147 | 27.4 |
| ≥ 21 potential PTSD | 70 | 13.0 |
| Total | 537 | 100.0 |

frequent PTSD symptom in the intrusion B category symptoms, symptom number 1 '*I try to stay away from people, places or things that remind me about what happened or what is still happening*' is the most frequent PTSD symptom in the avoidance C category symptoms, symptom number 5 '*I have trouble feeling happiness or love*' is the most frequent symptom of potential PTSD in Negative conditions/mood D category symptoms and symptom number 2 '*I get upset easily, or get into arguments, or physical fights*' is the most frequent PTSD symptom in the Arousal/Reactivity E category symptoms (S3–S6 Tables).

Univariate analysis with comparisons of the studied variables and risk factors for PTSD in the initial set of COVID-19 exposure questions revealed that work of a close relative around people who might have COVID-19 and upsetting issues that happened to children/adolescents or their families because of SARS-CoV-2 were significantly different between the groups of no, minimal, mild and potential PTSD symptoms ($X^2 = 14.7$, p = 0.002 and $X^2 = 12.2$, p = 0.007 respectively) but these significant differences disappeared during multivariate regression analysis comparing potential PTSD group to the other 3 groups with no, minimal and mild PTSD symptoms (S7 and S8 Tables).

Saudi children/adolescents had significantly lower median total UCLA BSCCA PTSD score than non-Saudi children/adolescents (H = 9.41, p = 0.002) with no other significant differences detected for other studied variables. Comparisons of total UCLA BSCCA PTSD scale score by nationality, region, age group, gender, and study level are presented in Table 5.

## Discussion

The global COVID-19 pandemic has significant impacts and serious adverse effects on the physical and mental health of people [10, 20]. Mental health affection by serious infectious pandemic threats has been recognized as an important public health challenge since a long time [21].

**Table 5. Comparisons of total PTSD UCLA brief scale score by nationality, region, age group, gender, and study level.**

| Variable | Categories Median, interquartile range (Q25-Q75) | | | | | H* df P |
|---|---|---|---|---|---|---|
| Nationality | Saudi (n = 494) | Non-Saudi (n = 43) | | | | 9.4 1.0 0.002 |
| | 7.0 (2.0–14.0) | 13.0 (7.0–23.0) | | | | |
| Region | Central (n = 48) | Western (n = 433) | Eastern (n = 19) | Northern (n = 23) | Southern (n = 14) | 4.7 4.0 0.31 |
| | 9.0 (3.0–21.0) | 8.0 (2.5–15.0) | 3.0 (1.0–9.0) | 6.0 (3.0–17.0) | 6.0 (3.0–13.25) | |
| Age group | 7–12 years (n = 276) | 13–18 years (n = 261) | | | | 0.06 1.0 0.80 |
| | 8.0 (2.0–15.0) | 8.0 (3.0–14.0) | | | | |
| Gender | Male (n = 262) | Female (n = 275) | | | | 1.12 1.0 0.30 |
| | 7.0 (2.0–15.0) | 9.0 (3.0–15.0) | | | | |
| Study level | Primary (n = 259) | Intermediate (n = 120) | Secondary (n = 158) | | | 0.09 2.0 0.96 |
| | 8.0 (2.0–16.0) | 7.0 (3.0–14.0) | 8 (3.0–14.0) | | | |

*Kruskal-Wallis H test, df: degree of freedom.

Cumulative sufficient evidences demonstrated that the extremely rapidly spreading COVID-19 global pandemic is a life-threatening infection that is serious enough to cause PTSD [20]. Most studies have focused on PTSD related to serious infectious diseases in the medical staffs [22]. PTSD among children/adolescents has been studied less frequently despite the fact that the critically developing children/adolescents are more vulnerable to psychological disturbances owing to their less mature cognitive abilities and adaptive capacities [23].

The COVID-19 pandemic has emerged as an unexpected disaster currently affecting essentially every country. The closure of schools, parks, public places and malls, keeping children locked at home was implemented to prevent further SARS-CoV-2 transmission, but the psychologic consequences of lockdown measures on families and children well-being should be considered. Therefore, this cross-sectional survey study was undertaken to screen for symptoms of PTSD associated with COVID-19 and its resultant long duration of quarantine and lockdown measures in children/adolescent in Saudi Arabia.

In this survey, which was conducted in KSA after 2 months from start of quarantine for COVID-19 pandemic, the results showed that a significant proportion (71.5%) of the participants had PTSD symptoms while they were in quarantine, with 44.1% and 27.4% of participating children/adolescents experienced symptoms of minimal and mild PTSD respectively while potential PTSD that warrant further evaluation and assessment was identified in 13% of participating children/adolescents.

There are other studies documenting the prevalence of PTSD related to disasters secondary to widespread infections. One study utilized a closely similar measuring tool to (BCSCA-U-CLA) which is the 17-item self-report PTSD Checklist-Civilian Version to identify PTSD symptoms [20]. It showed that the prevalence of PTSD was 12.8% within one month after the outbreak of COVID-19 in China, but the majority of participants in that study were between 21 to 30 years of age (range was from 14 to 35 years). Thus, the population screened was older than in our study where participating children/adolescents had a mean age of 12.25±3.77 years and age range from 7–18 years. The recorded COVID-19-related potential PTSD prevalence in this survey was also lower than the recorded symptoms of PTSD prevalence of 28.9% in 129 participants (64% of them were adults between 26–45 years and 68% of them were healthcare workers) who were quarantined for even a shorter period with a median duration of 10 days in response to SARS-CoV-1 pandemic in Toronto, Canada [7]. Additionally, the recorded COVID-19-related potential PTSD prevalence in this survey was also lower than the prevalence of PTSD of 30% in children who experienced quarantine for H1N1 in 6 states of USA, Mexico and Canada that was detected by University of California at Los Angeles Posttraumatic Stress Disorder Reaction Index (PTSD-RI) [16]. The PTSD-RI is so closely related and more comprehensive than (BCSCA-UCLA). However, the (BCSCA-UCLA) has the advantages of being simple, brief, self-administered by child/adolescent and designed specifically for COVID-19 related PTSD. Thus, significant variations are expected in PTSD prevalence that may be due to the differences in the age and characters of participants including their possible underlying genetic and health conditions predisposing to PTSD, research methods, diagnostic criteria or measuring tools. Variations are also expected due to differences in cultures, severity and nature of the disaster and time period passed after the main traumatic event.

It is worth to mention that it is too difficult or impossible to disentangle the effects of pandemic COVID-19 versus the lockdown in this survey because our participants were experiencing and suffering from both the ongoing COVID-19 pandemic and lockdown during the time of the study. However, in a study to evaluate the psychological, behavioral, and psychiatric assessment of Saudi children exposed to the 2009–2010 South war in Jazan compared to children unexposed to war found that the prevalence of PTSD in unexposed children was only 1.7% [24]. So, there is a remarkable significant rise in the prevalence of PTSD from a

pre-pandemic level of 1.7% to 13% in children/adolescents during COVID-19 pandemic and lockdown.

In the present study, we tried to investigate the risk factors associated with potential PTSD in Saudi children/adolescents by univariate and multivariate analysis of the demographic characters and the most important traumatic events of COVID-19. The current study examined characteristics of participants such as age, school grade, gender, region of residence in KSA and family size to be predictive or associated with PTSD symptoms, yet the results were negative. In a previous research, the results were different, and it was found that younger age was predictive of disaster-induced PTSD [25]. However, younger children may be less able to recognize and estimate the dangers and consequences of COVID-19 and consequently suffer less PTSD than older children. Similarly, another study found that age has no significant association with PTSD [7].

Women have been experiencing double the rates of PTSD compared to men, highlighting the differences due to gender. Adolescence is the expected period for the emergence of these gender differences in symptoms and estrogen is likely the reason due to its effects on neurobiology, leading to increased risk for PTSD in girls [26]. On the contrary, in the present study, there is no significant gender variation in symptoms of PTSD between boys and girls, most probably because nearly 50% of participants were within the pre-pubertal age group (7–12 years). This is similar to a study that found no gender differences in PTSD symptoms in children between 7 and 11 years old [27].

In this study, work of a close relative around people who might have COVID-19 (29.8% of participants) and upsetting issues that happened to children/adolescents or their families because of COVID-19 (9.5% of participants) were significantly different between the groups of no, minimal, mild and potential PTSD symptoms but these significant differences disappeared during multivariate regression analysis. Similarly, it was reported that the impact on children of parents who are frontline fighters is different as compared to parents doing most of their work from home during the pandemic. Quarantine/home confinement is an excellent opportunity for parents-children interaction, but the parents who are providing continuous health services are tired, over pressured and can't find enough time for their children. These children are missing their fathers and mothers due to extended periods of distancing [28].

In the current study, Saudi children/adolescents had significantly lower median total UCLA BSCCA PTSD score than non-Saudi children/adolescents. This can be explained by the impact of difficulties encountered by expatriate families to adjust to multiple challenges and stressors such as living in a non-native environment, financial constraints, need to readjust life in a new country including learning local language and culture, new schooling system, adapt to a new work and change of social environment, sense of uncertainty and displacement/isolation affecting all family members [29]. This is a unique feature and important finding in our study because up to our knowledge, no previous studies have addressed the comparison of COVID-19 related PTSD or even PTSD between children of citizens of the country and children of expatriate families.

In this study, participants declared that symptoms number 4, 1, 5 and 2 were the most frequent potential PTSD symptoms in the intrusion B category symptoms, the avoidance C category symptoms, the native conditions/mood D category symptoms, and in the arousal/reactivity E category symptoms. As expected, these 4 symptoms were present at a higher percentage in potential PTSD group compared to no, minimal and mild PTSD groups. Consequently, these simple 4 symptoms can be selected to draw the attention of parents and healthcare providers to suspect or notice the possible early evolution of PTSD in traumatized children/adolescents as soon as possible to provide them with proper timely intervention before progression into florid PTSD. Additionally, it should be realized that COVID-19 is

going hand in hand with a pandemic of childhood mental illnesses including depression, anxiety and pervasive developmental disorders and childhood obsession [28].

The present study has multiple strengths. Up to our knowledge, it is the first cross-sectional study in KSA and in the Middle East for screening of COVID-19 related PTSD in both children/adolescents of Saudi families and children/adolescents of expatriate non-Saudi families. Additionally, this study is characterized by the adequate sample size with high power of the study (more than 99%), the inclusion of all 5 nationwide regions of KSA, and the use of the UCLA-BCSCA which is a validated, simple and brief measuring tool that is specifically designed for COVID-19 related PTSD and can be used by older children and adolescents themselves or with the help of their parents.

However, this study is not without limitations because there was a higher participation from the western and central regions and lower participation from eastern, northern and southern regions of the kingdom. This can mainly be attributed to the presence of the higher population densities in these 2 regions as well as the uneven distribution of the online questionnaire, which depended on social media and internet resources/accessibility. However, this online questionnaire was the only way to reach participants in view of the inability to directly approach participants in different regions due to lockdown measures. Other limitations may be related to the use of a questionnaire in two different languages and adopting both parent and self-rating with the potential of influencing the findings of this study. Moreover, the UCLA-BCSCA is considered only as a screening tool to detect mainly potential PTSD that warrant further in-depth assessment but follow up of children/adolescents who had potential PTSD to confirm and manage PTSD could not be done. Moreover, other characteristics of participants especially their underlying genetic constitution and preexisting psychiatric or chronic illness disorders that may influence the development of PTSD could not be identified and need further in-depth analysis by another detailed study.

## Conclusions

In this study, COVID-19 pandemic and its resultant quarantine were associated with significant negative impact on psychological wellbeing of children and adolescents. Participating children/adolescents experienced symptoms of minimal, mild and potential PTSD in percentages of 44.1%, 27.4%, and 13% respectively. The characteristics of participants such as age, school grade, gender, region of residence in KSA and number of children in the family were not predictive or associated with PTSD symptoms. However, work of a close relative around people who might have COVID-19 and upsetting issues that happened to children/adolescents or their families because of SARS-CoV-2 were significantly associated with potential PTSD symptoms in univariate but not in multivariate regression analysis. Children and adolescents of expatriate families had higher total UCLA-BCSCA scores or more severe PTSD symptoms than children/adolescents of Saudi citizens. Specifically, 4 symptoms in the 4 main PTSD categories of intrusion, avoidance, negative mood and arousal/reactivity could be considered as alarming key symptoms that should alert parents and healthcare providers for early recognition of PTSD evolution in children/adolescents who need further in depth evaluation and management. This study highlighted the importance of COVID-19 related PTSD that should not be overlooked in different populations as it is expected in a significant proportion of children/adolescents with variable prevalence, risk factors and degree of severity. This study have public health implications with particular importance in clinical practice as parents, teachers and healthcare providers (pediatricians, psychologists, psychiatrists) must be aware of the psychological consequences of the COVID-19 pandemic including PTSD. So, they can be prepared with effective strategies/advice on how to handle the commonly expected PTSD during

the ongoing COVID-19 pandemic or similar disasters in future, providing children/adolescent with effective coping mechanisms.

## Supporting information

**S1 Fig. PASS output for calculation of the power of study.**
(DOCX)

**S1 Table. Upsetting issues associated with or caused by COVID-19.**
(DOCX)

**S2 Table. Frequency distribution of University of California at Los Angeles brief COVID-19 screen for child/adolescent PTSD questionnaire variables.**
(DOCX)

**S3 Table. Frequency distribution of intrusion category B symptoms in 4 PTSD categories.**
(DOCX)

**S4 Table. Frequency distribution of avoidance category C symptoms in 4 PTSD categories.**
(DOCX)

**S5 Table. Frequency distribution of negative conditions/mood category D symptoms in 4 PTSD categories.**
(DOCX)

**S6 Table. Frequency distribution of arousal/reactivity symptoms category E symptoms in 4 PTSD categories.**
(DOCX)

**S7 Table. Univariate analysis of risk factors between categories of PTSD.**
(DOCX)

**S8 Table. Multinomial regression for all studied variables or risk factors associated with potential PTSD compared to the other 3 combined groups (group without PTSD, group with minimal and group with mild PTSD).**
(DOCX)

## Author Contributions

**Conceptualization:** Mohamed H. Sayed, Moustafa A. Hegazi.

**Data curation:** Mohamed H. Sayed, Moustafa A. Hegazi.

**Formal analysis:** Mohamed H. Sayed, Moustafa A. Hegazi, Mohamed S. El-Baz, Turki S. Alahmadi, Nadeem A. Zubairi, Mohammad A. Altuwiriqi, Fajr A. Saeedi, Ali F. Atwah, Nada M. Abdulhaq, Saleh H. Almurashi.

**Investigation:** Mohamed H. Sayed, Moustafa A. Hegazi, Mohamed S. El-Baz, Turki S. Alahmadi, Nadeem A. Zubairi, Mohammad A. Altuwiriqi, Fajr A. Saeedi, Ali F. Atwah, Nada M. Abdulhaq, Saleh H. Almurashi.

**Methodology:** Mohamed H. Sayed, Moustafa A. Hegazi, Mohamed S. El-Baz, Turki S. Alahmadi, Nadeem A. Zubairi, Mohammad A. Altuwiriqi, Fajr A. Saeedi, Ali F. Atwah, Nada M. Abdulhaq, Saleh H. Almurashi.

**Supervision:** Mohamed H. Sayed, Moustafa A. Hegazi.

**Validation:** Mohamed H. Sayed, Moustafa A. Hegazi, Mohamed S. El-Baz, Turki S. Alahmadi, Nadeem A. Zubairi, Mohammad A. Altuwiriqi, Fajr A. Saeedi, Ali F. Atwah, Nada M. Abdulhaq, Saleh H. Almurashi.

**Writing – original draft:** Mohamed H. Sayed, Moustafa A. Hegazi, Mohamed S. El-Baz, Turki S. Alahmadi, Nadeem A. Zubairi, Mohammad A. Altuwiriqi, Fajr A. Saeedi, Ali F. Atwah, Nada M. Abdulhaq, Saleh H. Almurashi.

**Writing – review & editing:** Mohamed H. Sayed, Moustafa A. Hegazi, Mohamed S. El-Baz, Turki S. Alahmadi, Nadeem A. Zubairi, Mohammad A. Altuwiriqi, Fajr A. Saeedi, Ali F. Atwah, Nada M. Abdulhaq, Saleh H. Almurashi.

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
