## [Decision Letter · Decision Letter 0]

19 Apr 2021

PONE-D-21-07795

COVID-19 related posttraumatic stress disorder in children and adolescents in Saudi Arabia

PLOS ONE

Dear Dr. Hegazi,

Thank you for submitting your manuscript to PLOS ONE. After careful consideration, we feel that it has merit but does not fully meet PLOS ONE’s publication criteria as it currently stands. Therefore, we invite you to submit a revised version of the manuscript that addresses the points raised during the review process.

We look forward to receiving your revised manuscript.

Kind regards,

Vedat Sar, M.D.

Academic Editor

PLOS ONE

Journal Requirements:

2. Please change "female” or "male" to "woman” or "man" as appropriate, when used as a noun (see for instance https://apastyle.apa.org/style-grammar-guidelines/bias-free-language/gender).

3. In order to improve reporting, in your methods section, please provide additional information about the participant recruitment method and the demographic details of your participants, including a description of how and from where participants were recruited.

Reviewers' comments:

Reviewer's Responses to Questions

**Comments to the Author**

1. Is the manuscript technically sound, and do the data support the conclusions?

Reviewer #1: Yes

2. Has the statistical analysis been performed appropriately and rigorously? 

Reviewer #1: No

3. Have the authors made all data underlying the findings in their manuscript fully available?

Reviewer #1: No

4. Is the manuscript presented in an intelligible fashion and written in standard English?

Reviewer #1: Yes

5. Review Comments to the Author

Reviewer #1: The authors present a well powered cohort of over 500 indiviudals screened with the Brief COVID-19 Screen for

Child/Adolescent PTSD to test if the COVID-19 related lockdown increases risk for PTSD in children.

While the overall study is interesting, I have some open issues to be considered before publication.

Introduction:

The authors mention previous studies focusing on pandemics and PTSD, specifically, How does it relate to the previous study from UCLA

Also with regards to the discussion the validity of the screening instrument should be considered. i.e. is it adequate to form these subgroups or not.

For example the authros state the “Few studies [7, 16] have evaluated PTSD associated with infectious disasters” It would be intersting to report the main outomces of the studies and introduce them, to better understand what was the a-priori hypothesis was, and if and why the authors would expect differences in KSA cohorts

Data acquisition:

Was the screening filled by parents or children, i.e. is it self-report or not.

In the methods the authors state that “participants under 18 years old” participated. But in the results the authors talk about parents only. I did thus not understand if only parents filled the questionnaire, and if so, if the father or the mother evaluated the child’s behaviour.

I am not a big fan of including more than one individual per family, as the environmental conditions for each individual are presumably the same and thus the effects get overestimated. Where the authors able to consider this, or how was this accounted for.

Why was age considered as binary parameter and not as a quantitative parameters

How was the translation done and how was the translation evaluated. Information can get lost during translation of screening questionnaires.

Results:

Please report effectsizes whenever possible. In the statistics section the authros report to test median group differences using KW-test, but report an F-value rather the H-value without the deegrees of freedom. In case of significant effects please also provide pairwise comparisons.

Further It is not clear to me why the main readout of the PTSD Screening questionnaire was reported as mulinomial problem only and not as a quantitative regression problem.

In regression analysis also provide the overall model significance

Was multicollinearity between the predictors investigated

Discussion

In the discussion PTSD diagnosis, PTSD symptoms present not Present and meeting the PTSD screening cutoff are mixed, please adjust accordingly. these are three different measures.

The authors report that the identified PTSD rate of 13% is lower than expected, however to my opinion it is pretty similar to the 12.8% identified within one month after the outbreak of COVID-19 in China.

Indeed it is however lower than the PTSD of 30% in children who experienced quarantine for H1N1 in 6 states of USA, Mexico and Canada [16].

A critical review of the different tools and measures within the studies would be highly recommended. Also it would be interesting to see if the rate in the expats cohorts might be similar.

I am not sure if the presented study can make a conclusion on the pandemic versus the lockdown. It would be necessary to disentangle these two dimensions at least by comparing the numbers to the frequencies of PTSD symptoms in a general pre-pandemic population.

Finally, while I agree that the current situation challenges resilience mechanisms of children, I would encourage to dig deeper into the data to understand if the individuals with parents and or siblings at home are more resilient compared to those where children are home alone during lockdown.

Also I would encourage to build a more detailed quantitative regression model predicting number symptoms.

Minor comments

Report in the abstract if the assessment is self or parental report

First the nomenclature between Sars-Cov2 (refereeing to the virus) and COVID (refereeing to the disease resulting from the virus) and the COVID19-Pandemic is not consistent and should be reviewed

provide the raw data

6. PLOS authors have the option to publish the peer review history of their article (what does this mean?). If published, this will include your full peer review and any attached files.

Reviewer #1: No

---

## [Author Response · Author response to Decision Letter 0]

10 May 2021

Dear Editor in Chief and Academic Editor of PLOS ONE Journal,

Thank you for the revision of this original article (PONE-D-21-07795: COVID-19 related posttraumatic stress disorder in children and adolescents in Saudi Arabia) and giving us the chance to respond and explain some issues that can answer all concerns of editor/reviewer (s). 

We have responded to all comments and each point raised by the academic editor and reviewer (s) and all required necessary changes were added to the manuscript appropriately at the relevant sites highlighted in red color and within the requested time frame for revision. 

Both a marked-up copy of the manuscript with highlighted changes (Revised Manuscript with Track Changes) and an unmarked version of the revised paper without tracked changes labelled (Manuscript) were uploaded as 2 separate files. 

Reply to Comments of the Academic Editor:

1. The PLOS ONE's style requirements for the manuscript meets, including those for file naming were met. 

2. "Male” or "Female" were changed to "Boy” or "Girl" as appropriate when used as a noun but not to “Man” or “Woman” because the participants were children and adolescents. 

3. The request of additional information about the participant recruitment method and the demographic details of participants, including a description of how and from where participants were recruited. 

Response/Reply:

The participant recruitment method, including a description of how and from where participants were recruited, was described in the materials and methods section of the originally submitted manuscript under: 

A. The subheading (Study design and selection of participants) in page 12, lines 109-112:

Parents of children and adolescents were randomly selected and approached by an electronic online form of BCSCA. Parents were invited to answer this questionnaire if they were Saudi citizens or residents and had children/adolescents with age range of 6-18 years, who experienced the COVID-9 quarantine.

B. The subheading (Questionnaire implementation and distribution) in page 14, lines 146-150:

Both English and Arabic questionnaires were converted to Google forms, so participants can select to fill the most convenient questionnaire for them. Then, the links of both questionnaires were sent via social media, including WhatsApp’s, Facebook and Twitter, to participants. Participants were allowed to send the questionnaire’s link to their relatives and friends as a mean to increase the sample size.

Regarding the demographic details of participants, we included (Nationality, Region in KSA, Number of children in family, Age and gender of child/adolescent, School grade of child/adolescent) as in the original Brief COVID-19 Screen for Child/Adolescent (BCSCA) PTSD developed by University of California at Los Angeles UCLA (BCSCA-UCLA). An initial thinking to include more demographic details was cancelled because the questionnaire will be lengthier and participants may not be encouraged or refuse to answer such long questionnaire with inquiry about more private details of their life as monthly family income, level of education and occupation of mothers and fathers and type of home (rent or own apartment/villa). The idea was to encourage as much participants as possible to answer the main relevant questions of the (BCSCA-UCLA) which increases the sample size and power of the study. These main relevant questions have a priority and are much more important than questions for other demographic details of participants because they concentrate on risk factors of COVID-19 related PTSD such as affection of children/adolescents or their families and near relatives/friends by the COVID-19 and its serious effects to be followed by questions about the frequency of PTSD symptoms. Thus, there is an initial set of 7 questions to briefly review the traumatic event, assist the child in recalling details of the traumatic event and set the stage for the subsequent 11-item set of validated questions about the frequency of PTSD symptoms in the past month. Therefore, we did not want to make a longer questionnaire with including relatively less important questions about other demographic details of participants which may result in reluctance and unwilling of the participants to share or complete the provided (BCSCA-UCLA) questionnaire. 

Reply to Comments of the Reviewer (s):

1. The reviewer was concerned about making all data underlying the findings in the manuscript fully available. 

Response/Reply: The data was provided as part of the manuscript and in the supplementary materials or supporting information. Additionally, the raw data SPPS file was made available and deposited to figshare public data repository. DOI: 10.6084/m9.figshare.14565531

2. In the introduction, the reviewer asked about previous studies focusing on pandemics and PTSD, specifically, and how does it relate to the previous study from UCLA. 

Response/Reply:

In the introduction, we have just mentioned 2 studies (References number 7, 16) which evaluated PTSD associated with infectious pandemic disasters in children/adolescents but we did not mention any relation to (BCSCA-UCLA) questionnaire because (BCSCA-UCLA) was presented later on in the material and methods section. However, the study of PTSD in children who experienced quarantine for H1N1 in 6 states of USA, Mexico and Canada (Reference number 16) utilized University of California at Los Angeles Posttraumatic Stress Disorder Reaction Index (PTSD-RI) which is so closely related and more comprehensive than (BCSCA-UCLA). The (BCSCA-UCLA) has the advantages of being simple, brief, self-administered by child/adolescent and designed specifically for COVID-19 related PTSD. This was added in the discussion section to compare between PTSD associated with infectious pandemic disasters in children/adolescents in Saudi Arabia and in other countries. Moreover, another recent Chinese study utilized a closely similar measuring tool (17-item self-report PTSD Checklist-Civilian Version) to identify PTSD symptoms in youth (Reference number 20). This was mentioned in discussion section of the originally submitted manuscript, page 22, lines 269-270. However, this was more emphasized and relation to (BCSCA-UCLA) was clearly mentioned. 

3. In the introduction, the reviewer mentioned that it would be interesting to report the main outcomes of the studies about PTSD associated with infectious pandemic disasters in children/adolescents and introduce them, to better understand what was the a-priori hypothesis was, and if and why the authors would expect differences in KSA cohorts. 

Response/Reply:

The main outcomes of these studies and the differences from our KSA cohort have been mentioned in details in the discussion section of the originally submitted manuscript, page 22, lines 268-286. This part may be more convenient to be mentioned in the discussion section while comparing outcomes of our study to outcomes of other similar studies.

4. In data acquisition, the reviewer inquired if the screening questionnaire was filled by parents or children, i.e. is it self-report or not. In the methods the authors state that “participants under 18 years old” participated. But in the results the authors talk about parents only. I did thus not understand if only parents filled the questionnaire, and if so, if the father or the mother evaluated the child’s behavior.

Response/Reply:

The screening BCSCA-UCLA questionnaire was filled by both parents and children as it was fully explained and mentioned in the materials and methods section of the originally submitted manuscript, page 14, lines 151-156 that: 

The questionnaire was directed to parents of children who can fill the whole questionnaire including response to questions directed for their children after taking opinions and answers first from their eldest child. Furthermore, the parent could also enter the questionnaire link again to fill further questionnaire(s) for other children between 6-18 years. The child/adolescent was also allowed to directly answer questions if desired, and he/she understood the questions with or without the help of his/her parents.

In the results section, parents were only mentioned because they either filled the questionnaire for their younger children after taking their responses (opinions/answers) to the included questions (Parental-reported/completed questionnaire) or provide the questionnaire for older children/adolescents who can understand and answer questions without any help (Self-completed questionnaire). However, if older children/adolescents ask for the help of their parents to explain any question, parents were present beside them as interviewers to explain any raised question or issue. So, parents were mainly mentioned in the results because they were pivotal in the collection of data for this questionnaire.

However, more clarification of the method of filling the questionnaire was included in the material and methods section under the subheading; Questionnaire implementation and distribution and parents were replaced by participating children/adolescents in results section..

5. In data acquisition, the reviewer mentioned: I am not a big fan of including more than one individual per family, as the environmental conditions for each individual are presumably the same and thus the effects get overestimated. 

Response/Reply:

Thanks to the reviewer for raining this issue. We agree that including more than one individual per family, as the environmental conditions for each individual may be presumably the same and thus the effects get overestimated. However, there are individual factors that may be more important than the environmental conditions (which will be constant factors for individuals of the same family) in predisposition for PTSD such as the age/study level which is closely related to developmental maturity level and ability to perceive the traumatic effects of COVID-19 resulting in PTSD. Additionally, gender and other individual risk factors are important risk factors associated with COVID-19 PTSD as mentioned in previous studies and discussed in the discussion section. 

6. In data acquisition, the reviewer inquired why was age considered as binary parameter and not as a quantitative parameter. 

Response/Reply:

This was done because we want to detect the difference in PTSD UCLA brief scale score between school children (7-12 years) and older adolescents (13-18 years) to have 2 main groups with considerable number in each group to get robust results when comparing such 2 groups with evident difference in age group and level of developmental maturity (i.e. there may be no marked difference in COVID-19 related PTSD between 7 and 8-year old child or between 13 and 14-year adolescent but considerable differences are expected when comparing between a group of school children and older adolescents). 

7. In data acquisition, the reviewer inquired about how was the translation done and how was the translation evaluated. 

Response/Reply:

It was mentioned in the materials and methods section of the originally submitted manuscript under the subheading; Questionnaire implementation and distribution, page 14, lines 143-145, that: The UCLA-BCSCA is originally available in English and it was translated into Arabic, checked by two bilingual experts and used in a pilot study for Arabic speaking participants to detect if any amendments are required.

8. In results, the reviewer mentioned that authors reported to test median group differences using KW-test, but report an F-value rather the H-value without the degrees of freedom. 

Response/Reply:

Thanks for the review for detecting this point. F-value was a typographic error because it is actually and certainly the H-value and not F-value. This typographic error was corrected and degrees of freedom were added. 

9. In results, the reviewer inquired why the main readout of the PTSD Screening questionnaire was reported as multinomial problem only and not as a quantitative regression problem.

Response/Reply:

This was related to the type of the studied variables included in the analysis or comparisons such as nationality (whether Saudi citizen or Non-Saudi resident), region (whether participants were from central, eastern, western, northern or southern region), age group (7-12 year old school children versus 13-18 year older adolescents), gender (either boy or girl) and study level (primary versus intermediate versus secondary school level). All the previously mentioned variables are represented in a qualitative way (whether present (Yes) or absent (No) and not in quantitative way in the form of (mean or median, standard deviation and range).

10. In regression analysis, the reviewer requested to provide the overall model significance

and asked if multicollinearity between the predictors were investigated. 

Response/Reply:

The overall model significance of multivariate regression analysis wasn’t provided and multicollinearity between the predictors were not investigated because the regression model did not detect significant predictors associated with COVID-19 related PTSD. Only Saudi nationality persisted as significant factor with significantly more Saudi children with potential PTSD than non-Saudi children with potential PTSD (certainly number of Saudi participants were significantly more non-Saudi) but the total BCSCA scale score (i.e. severity of PTSD symptoms) was significantly higher in non-Saudi than Saudi children/adolescents. It was mentioned in the results section of the originally submitted manuscript, page 20, lines 234-236 and in Table S8 in the supplementary materials that: these significant differences disappeared during multivariate regression analysis comparing potential PTSD group to the other 3 groups with no, minimal and mild PTSD symptoms (Supplementary information, S8 Tables).

11. In the discussion, the reviewer mentioned that PTSD symptoms present/not present and meeting the PTSD screening cutoff are mixed, please adjust accordingly as these are three different measures.

Response/Reply:

This issue was explained and it was addressed in materials and methods section of the originally submitted manuscript, pages 13 and 14, lines: 137-141, that: The UCLA BCSCA assessment tool includes reaction index total scale score based on (DSM-5) PTSD diagnostic screener, with screener rating from 1-10 denoting minimal PTSD symptoms and rating from 10-20 denoting mild PTSD symptoms whereas rating of 21 or higher denotes potential PTSD and warrants further evaluation 140 by full PTSD-reaction index assessment and triage.

Accordingly, it was clearly mentioned in the discussion section of the originally submitted manuscript, pages 21 and 22, lines: 262-267, that: In this survey, which was conducted in KSA after 2 months from start of quarantine for COVID-19 pandemic, the results showed that a significant proportion (71.5%) of the participants had PTSD symptoms while they were in quarantine, with 44.1% and 27.4% of participating children/adolescents experienced symptoms of minimal and mild PTSD respectively while potential PTSD that warrant further evaluation and assessment was identified in 13% of participating children/adolescents 

12. In the discussion, the reviewer pointed out that the authors report that the identified PTSD rate of 13% is lower than expected; however to my opinion it is pretty similar to the 12.8% identified within one month after the outbreak of COVID-19 in China. 

Response/Reply:

Yes we completely agree with the reviewer on this point and we have discussed our results in comparison with this particular Chinese study in youth as it was mentioned in the discussion section of the originally submitted manuscript, pages 22, lines: 269-274, that: One study utilized a closely similar measuring tool (17-item270 self-report PTSD Checklist-Civilian Version) to identify PTSD symptoms [20]. It showed that the prevalence of PTSD was 12.8% within one month after the outbreak of COVID-19 in China, but the majority of participants in that study were between 21 to 30 years of age (range was from 14 to 35 years). Thus, the population screened was older than in our study where participating children/adolescents had a mean age of 12.25±3.77 years and age range from 7-18 years. 

13. In the discussion, the reviewer mentioned that indeed it is however lower than the PTSD of 30% in children who experienced quarantine for H1N1 in 6 states of USA, Mexico and Canada [16]. A critical review of the different tools and measures within the studies would be highly recommended. 

Response/Reply:

Yes we completely agree with the reviewer on this point and critical review of different tools/measures and characteristics of the participants was done and explanations for variations between outcomes of different studies were mentioned in the discussion section of the originally submitted manuscript, pages 22, lines: 281-286 as it was mentioned that: Thus, significant variations are expected in PTSD prevalence that may be due to the differences in the age and characters of participants including their possible underlying genetic and health conditions predisposing to PTSD, research methods, diagnostic criteria or measuring tool. Variations are also expected due to differences in cultures, severity and nature of the disaster and time period passed after the main traumatic event.

14. In discussion, the reviewer raised attention that also it would be interesting to see if the rate in the expats cohorts might be similar. 

Response/Reply:

Thanks too much for the reviewer for raising this point as children/adolescents of expatriate families had significantly higher median total PTSD score than children/adolescents of the Saudi citizens. This is a point of strength, a unique feature and important finding in our study because up to our knowledge, no previous studies have addressed the comparison of COVID-19 related PTSD or even PTSD between children of citizens of the country and children of expatriate families. 

15. In the discussion, the reviewer said that I am not sure if the presented study can make a conclusion on the pandemic versus the lockdown. It would be necessary to disentangle these two dimensions at least by comparing the numbers to the frequencies of PTSD symptoms in a general pre-pandemic population.

Response/Reply:

Thanks again for the reviewer for raising this issue. We completely agree with the reviewer that it is too difficult or impossible to make a conclusion on the pandemic versus the lockdown because our participants were experiencing and suffering from both the ongoing COVID-19 pandemic and lockdown during the time of the study. However, the reviewer promoted and alerted us to search for the frequencies of PTSD symptoms in a general pre-pandemic population. A study by El Hatw et al, 2015 to evaluate the psychological, behavioral, and psychiatric assessment of Saudi children exposed to the 2009-2010 South war in Jazan compared to children unexposed to war found that the prevalence of PTSD in unexposed children (general pre-pandemic population) was only 1.7%. 

El Hatw MM, El Taher AA, El Hamidi A, Alturkait FA. The association of exposure to the 2009 south war with the physical, psychological, and family well-being of Saudi children. Saudi Med J. 2015; 36 (1): 73–81. doi: 10.15537/smj.2015.1.9494

So, there is a remarkable significant rise in the prevalence of PTSD from 1.7% in general pre-pandemic population to 13% in Saudi children during COVID-19 pandemic and lockdown. This section was added and highlighted in the discussion section and the relevant reference was added. 

16. In the discussion, the reviewer encourages to dig deeper into the data to understand if the individuals with parents and or siblings at home are more resilient compared to those where children are home alone during lockdown. 

Response/Reply:

This was done (please see S7 Table: Univariate analysis of risk factors between categories of PTSD in the supplementary material). The number of children/family did not differ significantly between category groups of PTSD (X2=22.6, p= 0.09) in univariate regression analysis. The work of close relative around people who might have COVID-19 was significantly higher in children/adolescents with more severe (Potential) PTSD (X2=14.7, p= 0.002) in univariate regression analysis but this significant difference disappeared in multivariate regression analysis.

17. The reviewer encourages building a more detailed quantitative regression model predicting number symptoms. 

Response/Reply:

The qualitative regression analysis model was used as it was explained in response to point number 9 that: 

This was related to the type of the studied variables included in the analysis or comparisons such as nationality (whether Saudi citizen or Non-Saudi resident), region (whether participants were from central, eastern, western, northern or southern region), age group (7-12 year old school children versus 13-18 year older adolescents), gender (either boy or girl) and study level (primary versus intermediate versus secondary school level). All the previously mentioned variables are represented in a qualitative way (whether present (Yes) or absent (No) and not in quantitative way in the form of (mean or median, standard deviation and range).

18. In minor comments, the reviewer asked to report in the abstract if the assessment is self or parental report

Response/Reply:

This was reported and highlighted in the abstract as the questionnaire can be parent-reported or self-completed by older children/adolescents themselves.

19. In minor comments, the reviewer requested to review the nomenclature between Sars-Cov2 (refereeing to the virus) and COVID (refereeing to the disease resulting from the virus) and the COVID19-Pandemic as it is not consistent.

Response/Reply:

The nomenclature between Sars-Cov2 (refereeing to the virus) and COVID (refereeing to the disease resulting from the virus) and the COVID19-Pandemic is reviewed and it is consistent.

---

## [Decision Letter · Decision Letter 1]

17 Jun 2021

PONE-D-21-07795R1

COVID-19 related posttraumatic stress disorder in children and adolescents in Saudi Arabia

PLOS ONE

Dear Dr. Hegazi,

Thank you for submitting your manuscript to PLOS ONE. After careful consideration, we feel that it has merit but does not fully meet PLOS ONE’s publication criteria as it currently stands. Therefore, we invite you to submit a revised version of the manuscript that addresses the points raised during the review process.

We look forward to receiving your revised manuscript.

Kind regards,

Vedat Sar, M.D.

Academic Editor

PLOS ONE

Journal Requirements:

Reviewers' comments:

Reviewer's Responses to Questions

**Comments to the Author**

1. If the authors have adequately addressed your comments raised in a previous round of review and you feel that this manuscript is now acceptable for publication, you may indicate that here to bypass the “Comments to the Author” section, enter your conflict of interest statement in the “Confidential to Editor” section, and submit your "Accept" recommendation.

Reviewer #1: All comments have been addressed

2. Is the manuscript technically sound, and do the data support the conclusions?

Reviewer #1: Partly

3. Has the statistical analysis been performed appropriately and rigorously? 

Reviewer #1: Yes

4. Have the authors made all data underlying the findings in their manuscript fully available?

Reviewer #1: No

5. Is the manuscript presented in an intelligible fashion and written in standard English?

Reviewer #1: Yes

6. Review Comments to the Author

Reviewer #1: The authors have addressed the questions adequately

And the manuscript has significantly improved, however the question rgarding the linear regression model (as opposed to the multinominal model) was not answered appropriately. I suggest to adapt a linear regression model (preferebly with fixed AND random effect (reasons below) using PTSD symptom count as dependent variable. This has much more power to detect small effects and allows to adequately correct for hierarchical biases as introduced by rater and or language.

I still have some problem with using a questionnaire in two different languages: Translation and cultural adaption of a questionnaire can bias a result. Specifically, since translation of a questionnaire to my opinion is not trivial (i.e. forward and backward translations and validation of the psychometric constructs) . The authors should mention this in the limitation section or detail their efforts in the methods and supplements.

This is specifically relevant, since the main finding is within the subcohort of expats, where I would assume most of the English questionnaires have been collected.

This potential limitation and potential bias of the study should be clearly mentioned, or even better be tested in the paper.

The second limitation is the potential bias between self and parent rating.

Thus I again strongly suggest to include a sensitivity analysis to ensure that language and/or rater do not influence the finding.

Overall the main finding reported in the Paper is a comparison between groups without correcting for potential confounders. This might also explain why the multinominal association association is not replicating the frinding sfrom the KW test.

Data Sharing:

Overall the authors provided summary statistics in the supplement, which however does not allow replicating the findings.

If the original data can be downloaded somewhere for replication purpose, the authors should highlight this.

Very minor comment

During the revison process some minor typos have slipped in (e.g. line 101)

7. PLOS authors have the option to publish the peer review history of their article (what does this mean?). If published, this will include your full peer review and any attached files.

Reviewer #1: No

---

## [Author Response · Author response to Decision Letter 1]

27 Jun 2021

Dear Editor in Chief and Academic Editor of PLOS ONE Journal,

Thank you for the second revision of this original article (PONE-D-21-07795: COVID-19 related posttraumatic stress disorder in children and adolescents in Saudi Arabia) and giving us the chance to respond and reply to all concerns of the reviewer to provide the most accurate presentation of the results of this original research and significantly improving the written manuscript. 

We have responded to all comments and each point raised by the reviewer and all required necessary changes were added to the manuscript appropriately at the relevant sites highlighted in red color and within the requested time frame for revision. 

Both a marked-up copy of the manuscript with highlighted changes (Revised Manuscript with Track Changes) and an unmarked version of the revised paper without tracked changes labelled (Manuscript) were uploaded as 2 separate files. 

Reply to the Comments of the Reviewer:

1. Regarding the question about linear regression model (as opposed to the multinomial model), the reviewer suggested to adapt a linear regression model (preferably with fixed AND random effect using PTSD symptom count as dependent variable because this has much more power to detect small effects and allows to adequately correcting for hierarchical biases as introduced by rater and or language.

Response/Reply:

We have pointed out that this was related to the type of the studied variables included in the analysis or comparisons such as nationality (whether Saudi citizen or Non-Saudi resident), region (whether participants were from central, eastern, western, northern or southern region), age group (7-12 year old school children versus 13-18 year older adolescents), gender (either boy or girl) and study level (primary versus intermediate versus secondary school level). All the previously mentioned variables are represented in a qualitative way (whether present (Yes) or absent (No) and not in quantitative way in the form of (mean or median, standard deviation and range).

Additionally and unfortunately the dependent or outcome variable (PTSD symptoms count or scale score) in this study is not normally distributed and the problem is that the results of the parametric tests including t-test generally used in linear regression for analysis, will affect the significance and reliability of the regression equation and its parameters. 

Moreover, normality violation and use of non-normally distributed dependent variable in linear regression, will affect the estimates of the standard error and the confidence interval, and hence the significance of the risk factors (independent variables or predictors). 

Finally, linear regression techniques still can be used even if normality is violated using a dependent variable that is not distributed normally and linear regression remains a statistically sound technique in studies of large sample sizes which provides appropriate sample sizes (i.e., >3000) and this extremely large sample size was not available in this study (Li et al, 2012). 

Li X, Wong W, Lamoureux EL, Wong TY. Are linear regression techniques appropriate for analysis when the dependent (outcome) variable is not normally distributed? Invest Ophthalmol Vis Sci. 2012 May 1; 53(6):3082-3. doi: 10.1167/iovs.12-9967. 

2. Regarding using a questionnaire in two different languages as translation and cultural adaption of a questionnaire can bias a result. The authors should mention this in the limitation section or detail their efforts in the methods and supplements.

Response/Reply:

In the first revision, we confirmed that necessary measures were undertaken to avoid potential bias that may arise as a result of Arabic translation and cultural difference as it was mentioned in the materials and methods section of the originally submitted manuscript under the subheading; Questionnaire implementation and distribution, lines 143-145, that: The UCLA-BCSCA is originally available in English and it was translated into Arabic, checked by two bilingual experts and used in a pilot study for Arabic speaking participants to detect if any amendments are required. However, the potential limitation of using a questionnaire in two different languages as translation and cultural adaption of a questionnaire can bias a result was also clearly mentioned in the limitations of the study. 

3. Regarding the potential bias between self and parent rating to ensure that language and/or rater do not influence the finding.

Response/Reply:

The potential bias between self and parent rating was added to study limitations.

4. Regarding data sharing: The reviewer was concerned about making all data underlying the findings in the manuscript fully available. 

Response/Reply: The data was provided as part of the manuscript and in the supplementary materials or supporting information. Additionally, it was clearly mentioned in the first revision that the original raw data SPPS file was made available without restriction, and deposited to figshare public data repository. DOI: 10.6084/m9.figshare.14565531 where it can be downloaded for replication purpose. 

5. Very minor comment: During the revision process some minor typos have slipped in (e.g. line 101)

Response/reply: Line 101 was rechecked.

---

## [Decision Letter · Decision Letter 2]

19 Jul 2021

COVID-19 related posttraumatic stress disorder in children and adolescents in Saudi Arabia

PONE-D-21-07795R2

Dear Dr. Hegazi,

We’re pleased to inform you that your manuscript has been judged scientifically suitable for publication and will be formally accepted for publication once it meets all outstanding technical requirements.

Kind regards,

Vedat Sar, M.D.

Academic Editor

PLOS ONE

Additional Editor Comments (optional):

Reviewers' comments:

Reviewer's Responses to Questions

**Comments to the Author**

1. If the authors have adequately addressed your comments raised in a previous round of review and you feel that this manuscript is now acceptable for publication, you may indicate that here to bypass the “Comments to the Author” section, enter your conflict of interest statement in the “Confidential to Editor” section, and submit your "Accept" recommendation.

Reviewer #1: All comments have been addressed

2. Is the manuscript technically sound, and do the data support the conclusions?

Reviewer #1: Partly

3. Has the statistical analysis been performed appropriately and rigorously? 

Reviewer #1: Yes

4. Have the authors made all data underlying the findings in their manuscript fully available?

Reviewer #1: Yes

5. Is the manuscript presented in an intelligible fashion and written in standard English?

Reviewer #1: Yes

6. Review Comments to the Author

Reviewer #1: I do not have any further suggestions. All my previous comments have been adressed or answered, thus I suggested publication of the manuscript

7. PLOS authors have the option to publish the peer review history of their article (what does this mean?). If published, this will include your full peer review and any attached files.

Reviewer #1: No

---

## [Editor Report · Acceptance letter]

27 Jul 2021

PONE-D-21-07795R2 

COVID-19 related posttraumatic stress disorder in children and adolescents in Saudi Arabia. 

Dear Dr. Hegazi:

I'm pleased to inform you that your manuscript has been deemed suitable for publication in PLOS ONE. Congratulations! Your manuscript is now with our production department. 

Kind regards, 

on behalf of

Dr. Vedat Sar 

Academic Editor

PLOS ONE